# Engineering digitizer circuits for chemical and genetic screens in human cells

Nicole M. Wong[1], Elizabeth Frias[2], Frederic D. Sigoillot [2], Justin H. Letendre[1], Marc Hild [2✉] &
Wilson W. Wong [1✉]

Cell-based transcriptional reporters are invaluable in high-throughput compound and CRISPR
screens for identifying compounds or genes that can impact a pathway of interest. However,
many transcriptional reporters have weak activities and transient responses. This can result
in overlooking therapeutic targets and compounds that are difficult to detect, necessitating
the resource-consuming process of running multiple screens at various timepoints. Here, we
present RADAR, a digitizer circuit for amplifying reporter activity and retaining memory of
pathway activation. Reporting on the AP-1 pathway, our circuit identifies compounds with
known activity against PKC-related pathways and shows an enhanced dynamic range with
improved sensitivity compared to a classical reporter in compound screens. In the first
genome-wide pooled CRISPR screen for the AP-1 pathway, RADAR identifies canonical genes
from the MAPK and PKC pathways, as well as non-canonical regulators. Thus, our scalable
system highlights the benefit and versatility of using genetic circuits in large-scale cell-based
screening.

[1] Department of Biomedical Engineering and Biological Design Center, Boston University, Boston, MA 02215, USA. [2] Novartis Institutes for BioMedical
Research, Cambridge, MA, USA. ✉email: marc.hild@novartis.com; wilwong@bu.edu

Successful therapeutics development requires the ability to interrogate the underlying disease biology and identify drug candidates that can modulate the disease phenotype. Cell-based screening, such as pooled genome-wide CRISPR knockout and high-throughput chemical library screens (HTS), have become indispensable for uncovering important protein targets and drug candidates that can regulate the phenotype of interest. A critical step in a cell-based screen is the accurate and sensitive measurement of various cellular phenotypes or pathway activities after genetic or chemical perturbations. Indeed, many approaches (e.g., viability, morphology, reporter systems) have been developed to measure cell states or quantify various pathway activities[1] for cell-based screenings. High-content approaches, such as imaging cytometry and single-cell sequencing, have also been adapted with cell-based screens to provide a more comprehensive view of cell states after genetic perturbation[2,3]. However, many output measurements cannot discern nuanced perturbations, thus overlooking important drug and gene candidates, and the application of high-content approaches to large-scale screens also remains challenging. Therefore, while these approaches have proven useful, they also illustrate the need for the design and optimization of output quantification for cell-based screening experiments.

One of the most widely used and important phenotypic readout methods in cell-based screenings involves pathway-specific transcription reporters, which express reporter genes in response to pathway activation[4–7]. These reporters are cost-efficient to implement and highly scalable, making them an ideal tool for large-scale chemical and genetic screens[8–10]. However, transcription reporters also have several critical deficiencies that limit their efficiency in cell-based screening, one being their weak activity that limits them to only identifying compounds and genes with strong effects on the pathway. Furthermore, the readout attained by these reporters is often proportional to the intensity of pathway activation. This presents a challenge as ambiguous signals make it difficult to determine whether the pathway was active. Many pathway activities are also transient and therefore necessitate screening at multiple timepoints to optimally capture pathway activation. These extra steps drastically increase the amount of resources needed, particularly for screening experiments with a large compound library. Therefore, given the prominent role that transcription reporters play in cell-based screenings, it would be advantageous to develop a reporter that can amplify and remember outputs of pathway activity.

The critical sensor element in a transcription reporter system is the promoter. A pathway-specific promoter is typically engineered by placing corresponding transcription factor binding sites (responsive elements) upstream of a minimal promoter. However, only a few promoter optimization strategies are available (e.g., increasing the number of binding sites or using different minimal promoters[11,12]), with most activity improvements marginal. Furthermore, promoter engineering alone cannot generate memory. As such, a different approach is needed to improve transcription reporters for screening applications. One of the most effective approaches for amplifying reporter activity is through the engineering of a recombinase-based digitizer circuit[13,14]. Site-specific recombinases are DNA modifying enzymes that recognize specific short nucleotide sequences (recombinase binding sites) to either invert, excise, insert, or translocate the DNA between them, depending on the orientation of these sites[15,16]. In this circuit, recombinase expression is driven by a pathway-sensitive promoter, allowing it to edit the sequence proximal to a reporter gene and resulting in constitutive reporter expression. As long as the pathway-sensitive promoter can generate sufficient recombinase, reporter expression is high, thus converting a weak and graded input signal into a digital-like and robust output.

While the basic design for this circuit has been previously established, we show here that extensive optimization is needed to make it applicable to high-throughput screening[17]. Specifically, our circuit incorporates a split recombinase to reduce background activity, and targeted genome integration is required to impart a digital readout. Single-cell analysis of our circuit illustrates this digital behavior with a clear distinction between "on" and "off" reporter gene expression. These characteristics result in a reporter that acts like an analog-to-digital converter, retains memory, and has high sensitivity. We have termed this system "Recombinase-based Analog-to-DigitAl Reporter" (RADAR).

As a proof of principle, a RADAR circuit was designed to monitor the activator protein (AP-1) pathway, specifically through the induction of protein kinase C (PKC) signal transduction. The AP-1 transcription factor is involved in many critical cell functions, such as proliferation, differentiation, and apoptosis[18]. It has also been implicated in tumorigenesis, with an increase in activity associated with multiple tumor types[19]. While PKCs have been widely studied, their signaling remains complex due to their diverse coverage of biological pathways and their impact on AP-1 activity is also not fully understood[20]. As such, a genome-scale CRISPR screen of the PKC-induced AP-1 pathway would provide a much-needed comprehensive view of the signaling network, as such information is lacking.

To determine the versatility of the RADAR system, we evaluated our AP-1 RADAR system in both large-scale compound HTS and pooled CRISPR screens. Pathway-specific transcription reporters are often used in these formats of large-scale screens, and here we show that RADAR is able to provide enhanced reporter performance. In the compound HTS screens, on top of providing memory to the reporter system, RADAR also demonstrates a large dynamic range and low basal activity. Using RADAR in a pooled CRISPR screen, we identified many canonical MAPK and PKC-related genes, consistent with their function in the AP-1 pathway. Furthermore, the enhanced signal provided by the RADAR system also allows us to uncover several non-canonical AP-1 pathway genes. Our RADAR system would greatly facilitate the identification of drug targets and candidates that would previously have been missed because of poor reporter performance, thus accelerating drug development.

## Results

**RADAR circuit designs.** The RADAR circuit was designed to encompass two main components: recombinase and reporter gene expression cassettes. The expression of the recombinase is dependent upon a pathway-sensitive promoter. The reporter expression cassette consists of a constitutive promoter followed by a termination signal (STOP) flanked by recombinase sites and a reporter gene. When expressed, the recombinase excises the stop element between the recombination sites, allowing for constitutive reporter gene expression (Fig. 1a). As a proof of concept, an AP-1 pathway-sensitive promoter was used to drive expression of a FlpO recombinase, and both GFP and luciferase were included as the reporter genes to allow for flexibility in bulk or single-cell readout. Phorbol 12-myristate 13-acetate (PMA) was used to induce the AP-1 pathway.

The choice of recombinase was found to be important, as the more potent Cre recombinase led to the activation of reporter gene expression in the absence of any pathway agonist. Furthermore, we found that leaky expression of the recombinase during cell culture led to unwanted background reporter activity. To address these issues, 16 designs were evaluated, including using shRNAs and redesigning the inducible promoter to contain different minimal promoters, weaker kozak sequences, and upstream open reading frames (ORFs), or quantity of short

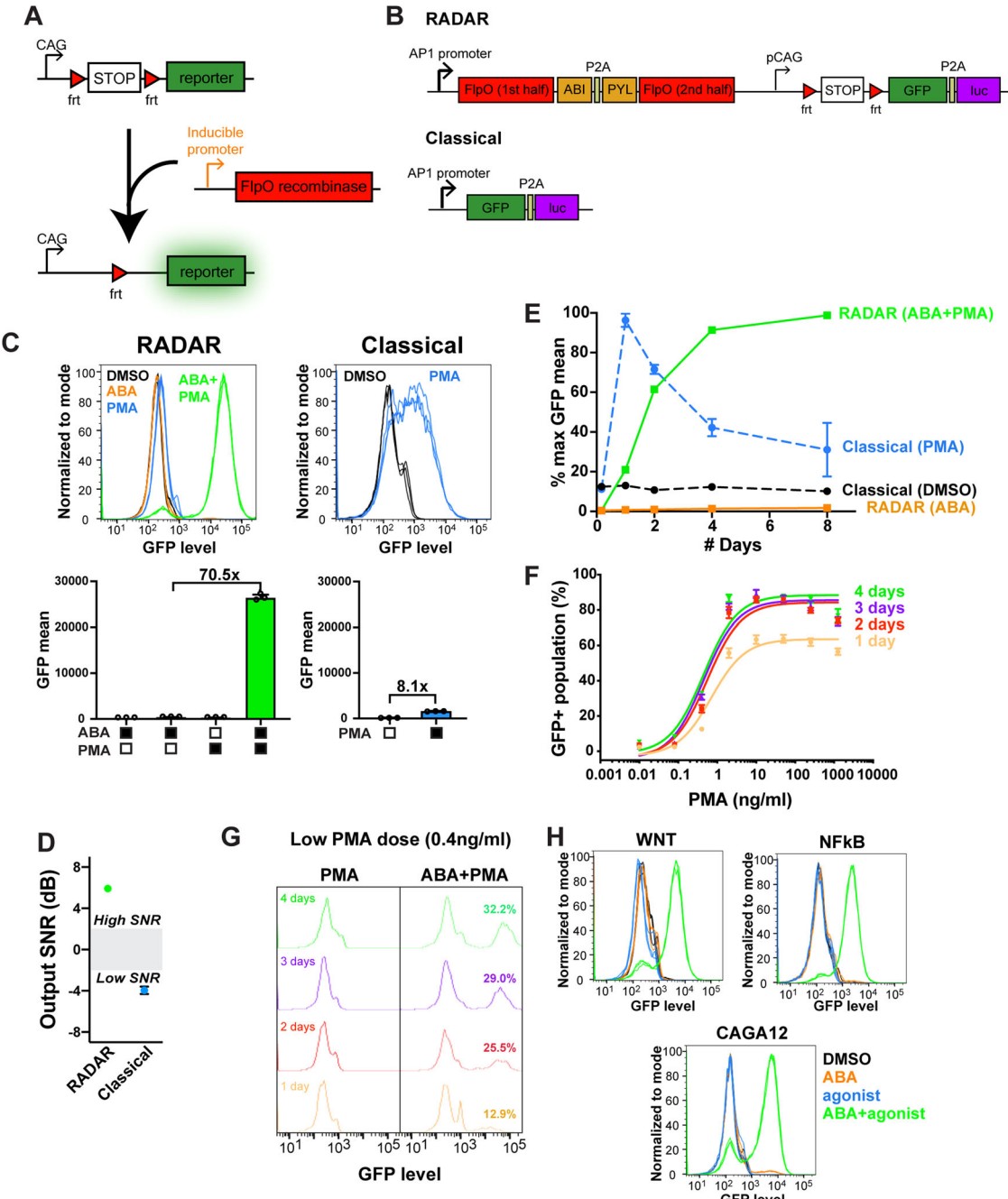

**Fig. 1 Development of RADAR. A** Schematic of how RADAR functions. The primarily tested inducible promoter in this study was sensitive to AP-1, though the system is also shown to be compatible with other promoters (WNT, NF-kB, or CAGA12). **B** Construct comparison between RADAR and a classical reporter. **C** Regulation of reporter-driven GFP expression using combinations of ABA, PMA, and DMSO (vehicle control) ($n = 3$, mean ± s.d.). **D** Signal-to-noise ratios of RADAR and classical reporters when comparing "on" (ABA/PMA for RADAR, PMA for classical) and "off" (ABA for RADAR, DMSO for classical) signals ($n = 3$, mean ± s.d.). **E** Retainment of GFP expression over time with comparison between RADAR and classical reporters ($n = 3$, mean ± s.d.). **F** Dose response of RADAR measured after 1, 2, 3, and 4 days' of incubation in ABA and PMA ($n = 3$, mean ± s.d.). **G** Sensitivity of RADAR to a low dose of PMA (0.4 ng/ml) over time with an increasing percentage of cells switching "on". **H** Application of RADAR to other pathway-sensitive promoters.

response elements and the spacers between them. Various degrons were also fused to the recombinase to lower basal levels. However, these approaches could not fully resolve the issue of background activity (Supplementary Fig. 1).

We found the most effective solution was to incorporate a split version of the recombinase. In this design, the two halves of FlpO were attached to a chemically induced proximity (CIP) system consisting of the domains ABI and PYL[14,21]. These domains are brought together by the plant hormone abscisic acid (ABA), thus forming a fully active enzyme and preventing any recombination of the reporter gene construct until the time of assay. While splitting the recombinase lowered basal activity, few cells remained switched on. In initial efforts, the reporter was tested in HEK293 cells after PiggyBac integration. Transient expression of the system indicated leaky behavior with ABA alone, likely due to the abundance of construct copies in the cell (Supplementary

Fig. 2a). As an alternative, the system was integrated through CRISPR/Cas9 knock-in at the AAVS1 safe harbor site. Integration at the AAVS1 site has been known to result in more consistent gene expression levels and a less variable cell population. As a result, we observed a dramatic shift in the number of cells switching on (Supplementary Fig. 2b). We further generated a monoclonal cell line incorporating this switch to ensure genome uniformity.

The resulting RADAR circuit was integrated into HEK293 cells together in a single construct (Fig. 1b). Various orderings and orientations of the cassettes were tested, with the most effective combination being the recombinase preceding the reporter. Whenever the first cassette was inverted, leaky reporter activity was observed in the presence of ABA alone, likely due to bleed over from the strong CAG promoter to the AP-1 promoter (Supplementary Fig. 2c). To ensure that expression of the recombinase was due to activation of the AP-1 promoter, the promoter was removed and the system became irresponsive in the presence of ABA and PMA (Supplementary Fig. 2d).

Reporters that are traditionally used in compound and CRISPR screens consist of a pathway-inducible promoter followed by a reporter gene. To evaluate how our system compares to the classical reporter, a classical AP-1 reporter was integrated into HEK293 cells in the same manner as the RADAR system. Results showed that our system achieved a fold change of 70.5 in GFP levels, whereas the classical system had fold changes of 8.1. Moreover, the shift in GFP median between "on" and "off" states observed by flow cytometry was more pronounced in RADAR and showed binary on/off behavior, suggesting that the agonist dose could be translated into a probability of flipping for our reporter (Fig. 1c). Quantification of luciferase activity showed similarly high levels in the "on" condition for RADAR, though the fold change observed was not as high (Supplementary Fig. 2e). For a more quantitative comparison, the signal-to-noise ratios (SNR) for both reporters were calculated[22], and it was found that while RADAR had a high SNR ($>6$ dB, over 2 dB is considered to be an excellent device), the classical reporter did not ($<-4$ dB) (Fig. 1d). Furthermore, RADAR only had a high SNR when both ABA and PMA were present (Supplementary Fig. 2f).

By permanently removing the termination signal using the recombinase, our system also allows for sustained "on" activity, thus providing memory, unlike the classical reporter. Both reporters were monitored for reporter gene expression over the course of 8 days, and it was verified that while the RADAR system maintained GFP expression over time, the classical reporter experienced a drop in GFP levels after the first day (Fig. 1e). In addition to the lack of memory, the degradation of PMA over time or natural attenuation of PKC signaling could potentially have led to this loss of GFP. While "on" activity of our reporter is maintained over time, the maximal percentage of cells that expressed GFP was achieved around 2 days after the addition of the agonist (Supplementary Fig. 3). This increase over 2 days may be due to the cumulative effect of recombinase activity within our system. The RADAR system further showed a dose response to PMA, with increased concentrations leading to higher recombinase protein expression, resulting in more cells switching "on". High concentrations of PMA resulted in "on" activity beginning to drop due to the toxicity of the drug killing the cells (Fig. 1f).

Longer incubation of our reporter with agonist also improved the sensitivity of the reporter. A low dose of 0.4 ng/ml PMA was tested in cells over time, and the percentage of cells switching "on" increased from 12.9 to 32.2% after 4 days (Fig. 1g), most likely reflecting the slow accumulation of recombinase and thus increasing the likelihood of switching the reporter "on". This suggests that our system may detect weaker activating drugs in compound screens by increasing the amount of time the cells are incubated with them.

To show that we can design RADAR for other pathways, we generated and tested WNT, NF-kB, and CAGA12 RADAR systems. All three systems showed a digital readout when both the pathway agonist (WNT-conditioned media, TNFα, and TGFß for the respective promoters) and ABA were present. Background levels in all three systems remained low if either ABA or the agonist was absent (Fig. 1h).

**Application to compound screening**. To examine how our cell-based reporter performs in a compound screen, it was tested against a library of 3494 compounds with high confidence annotations to $>2000$ different human targets, including >350 kinases, with eight different concentrations evaluated for each drug. This library is typically used to study compound mechanism of action (MoA) and can help identify genes that are involved in a pathway of interest. Others have used this well-defined library to study a variety of pathways, and findings from these screens are compiled for collaborative use[23]. We thus used this compound library to test whether drug hits that emerged would corroborate with our PKC-specific AP-1 pathway in both agonist and antagonist screens. In the agonist screen, HEK293 cells equipped with our RADAR system were treated with the compound library, incubated for 5 h, then stimulated with ABA to allow for the dimerization of recombinase domains. Cells were then incubated for two days, and a luciferase assay was performed. The antagonist screen was performed similarly, though stimulation was done using ABA and PMA to detect which drugs inhibited the AP-1 promoter (Fig. 2a). By first treating cells with the compound library, this allows for potential antagonists to inhibit the AP-1 pathway and prevent the later addition of PMA from activating the reporter. For comparison, the same screen was run using a classical AP-1 reporter, with cells treated with PMA or DMSO for screening agonists or antagonists, respectively, and an incubation period of 1 day.

To verify our reporter was correctly pulling out AP-1-related hits, we compared compounds within the MoA library that have been previously associated with the AP-1 pathway, PKC-related genes, or genes JUN or FOS (main subunits of the AP-1 transcription factor). Within the library, there were five agonists and 34 antagonists with previously established associations. The ability of the RADAR or classical reporter to identify these compounds as hits was then analyzed. Both reporters were able to identify four out of these five agonists, though RADAR showed a slight advantage by identifying 27 antagonists over the 24 found by the classical reporter (Supplementary Data 1 and 2). Results from screening these reporters against the MoA library thus corroborated with our pathway of interest.

Several other agonists were identified by both RADAR and the classical reporter without documented association to the PKC-induced AP-1 pathway (Supplementary Data 1). Upon a STRINGdb analysis of their known associated genes, we found a majority of them were linked to PKC or AP-1 through various pathways (Supplementary Fig. 4). Some of these compounds additionally indicated a clear dose response, similar to what was observed with previously established PKC-related agonists (Supplementary Fig. 5a). This suggests that these compounds may indeed regulate our pathway of interest and were not previously found in other MoA library screens. The antagonist screen yielded hundreds of other hits for both reporters, which was many more than the agonist screen. However, given the enrichment of known antagonists over agonists that have been previously established to be AP-1-related in the MoA library, this

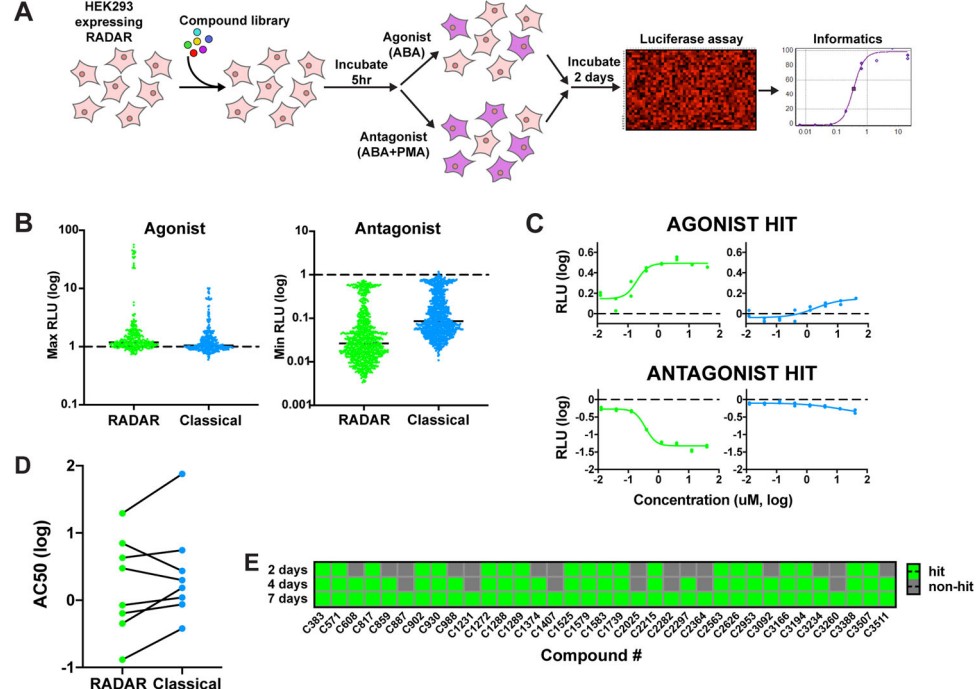

**Fig. 2 Application of RADAR to compound screening. A** Schematic of the compound screening process. **B** Maximum and minimum luciferase levels attained by each reporter for each compound in the agonist and antagonist reconfirmation screens, respectively. RLU values were normalized to the median of the negative controls. RADAR data is depicted in green, and classical reporter data depicted in blue. **C** Examples of compound hits found by RADAR that were missed by the classical reporter in agonist and antagonist reconfirmation screens (compounds C3166 and C3394, respectively). All RLU values were normalized to the median of the negative controls ($n = 2$). **D** AC50 values of shared agonist hits between both reporters in the reconfirmation screen. **E** Accumulation of specific compound hits in the agonist reconfirmation screen with 40,000 cells/ml cell density over time.

was not surprising. Several of these hits indicated a clear inhibitory dose response to increasing drug concentrations (Supplementary Fig. 5b). While many of these compounds have previously established associations with PKC and the MAPK pathway, the antagonist hits could also be causing toxicity and cell death, thus leading to a false positive for drug-induced inhibition of the AP-1 pathway (Supplementary Data 2). As such, it was difficult to determine which of these hundreds of compounds were real AP-1 antagonists with only a luciferase assay.

Reconfirmation screens were then performed to validate what was found in the initial MoA library screens. Due to the low number of agonist hits from the MoA library, the compounds that were included in the follow-up screen consisted of hits for RADAR, with the addition of drugs known to target the same pathways associated with them. Furthermore, known active and inactive analog pairs of the drug hits were included where possible to exclude the possibility of nonspecific activation. Drugs with similar properties to the initial hits, but different potencies (termed "range compounds"), were also included in the agonist reconfirmation screen. As the antagonist MoA library screen yielded several hundreds of hits, these hits were re-tested in the reconfirmation screen. Some hits from the agonist screen and compounds known to be inactive were included in the antagonist reconfirmation panel as negative controls. A total of 159 and 460 compounds were tested in the agonist and antagonist reconfirmation screens, respectively. In addition to the luciferase assay, cell viability assays were also performed in both reconfirmation screens to verify that changes in the bulk luciferase signal were not due to cell toxicity. This was particularly important for the antagonist screen to reduce false positives due to drug toxicity. To compare RADAR to the classical reporter, reconfirmation screens were performed with both reporters.

Visualizing the maximum luciferase readout attained for each compound, the digital nature of our reporter emerges. In the agonist screen, a subset of compounds was able to attain distinct signals from the rest of the panel. Compounds falling into this higher readout population in the agonist screen were enriched in hits from the initial MoA screen, range compounds associated with *CMA1* and *MAPK13*, and an active analog for an *EDNR*-related drug (Supplementary Data 3). The separation was less pronounced in the antagonist screen, though we still see pulling away from basal levels more than with the classical reporter. It was noted that both reporters had low basal activity in the agonist and antagonist reconfirmation screens (Fig. 2b).

As expected, likely due to activity from the strong CAG promoter, the dose–response curves obtained by our reporter had a larger dynamic range than those of the classical reporter. This was seen in both agonist and antagonist screens, allowing us to identify compound hits with RADAR that were missed with the classical reporter (Fig. 2c). Among the compounds identified by both reporters, we also found that agonist hits identified by both reporters were more potent with RADAR than the classical reporter (Figs. 2d and S6a). There were two compounds for which the AC50 value was lower for the classical reporter, but upon analysis of these agonists, we see that the dose–response curves for both reporters were actually comparable (Supplementary Fig. 6b). A lower AC50 indicates that our reporter has a higher sensitivity than the classical reporter and can detect lower drug doses.

We were curious about whether a longer incubation of the cell-based reporter with the drug would result in compounding of reporter readout, as was observed with the low dose of PMA tested previously. This could further increase the sensitivity of the assay and potentially pull out weaker agonists, so the duration between stimulation with ABA and luciferase assay readout was

varied to 2, 4, and 7 days and cell densities of 20,000, 40,000, 80,000, and 160,000 cells/ml were tested to factor in possible cell overgrowth. We found that the longer cells were incubated following stimulation, the more hits emerged from the screen. Increasing the cell density is also generally correlated with a larger number of compound hits. However, at the highest cell density after 7 days of incubation, the number of hits began to decrease, likely due to overgrowth of cells in the wells, limiting exposure of cells to the compounds (Supplementary Fig. 6c). To examine how the length of incubation impacted the compounds that were showing up as hits, a lower cell density of 40,000 cells/ml was focused on. Here, we found that the hits from 2 days of incubation were retrieved after 4 and 7 days of incubation, and the hits after 4 days carried through to 7 days as well (Fig. 2e). Many new hits emerged at the 4 and 7-day timepoints, and analysis of some of these dose–response curves showed a response to the drug after a longer incubation time, which could otherwise be missed (Supplementary Fig. 6d). These results, along with the lower AC50 values obtained by our reporter, suggest that RADAR not only has a higher sensitivity than the classical reporter but that the sensitivity can be further increased by lengthening the duration of compound incubation.

Comparing the reconfirmation screen results, there were several compound hits shared between the recombinase and classical reporters in both agonist and antagonist screens. A considerable overlap was observed between the hits for both reporters in the antagonist screen, but RADAR was able to identify more hits in the agonist screen (Supplementary Fig. 7a). In addition, comparing the hits detected by both reporters, we see a larger dynamic range obtained with RADAR (Supplementary Fig. 8b). Compound hits detected by only RADAR indicate a clear dose response, while hits detected by only the classical reporter do not (Supplementary Fig. 7c, d).

The antagonist screen results indicated several compounds found by both reporters that had previously been established to be inhibitors of PKC, cyclin-dependent kinases (CDKs), and MAPK proteins (Supplementary Data 4). As CDK and MAPK activity is known to be modulated by PKC, this finding was unsurprising. While some of the hits from the agonist screen also had known ties to our pathway of interest, associations with some of the other hits were not as straightforward. Several agonists found in the reconfirmation screen consisted of both active and inactive analogs targeting PCSK9. Upon closer inspection of these compounds, we see a clear dose response for some inactive analogs (Supplementary Fig. 8). However, as the results within active and inactive analog groups were inconsistent, the activity observed could be due to unselective compound behavior ("off-target" activity) and does not indicate a link between PCSK9 and our pathway. As previously described, our reporter had also pulled out potent agonists with known associations to *CMA1*, *MAPK13*, and *EDNR*. These compounds were additionally identified by the classical reporter as well (Supplementary Data 3). As MAPK13 is involved in the MAPK pathway, which is upstream of AP-1, this could account for its effect on our reporter[24]. PKC-induced AP-1 is also known to be downstream in the endothelin processing pathway and the angiotensin pathway, possibly explaining the affiliation with *EDNR* and *CMA1*, respectively[25,26]. Therefore, our screens provide further information on the downstream effects of these compounds.

As a control to verify that the hits observed in the reconfirmation screens were due to the activity of our recombinase, the agonist reconfirmation library was screened with our reporter without any ABA. In the absence of the small molecule, the two recombinase halves would not dimerize and remove the stop element in the reporter. Thus, any changes in luciferase levels would not be attributed to the regulation of our

AP-1 promoter. Results showed that no compound hits were found, indicating that the hits we were observing were due to expression of our AP-1-induced recombinase (Supplementary Fig. 9a).

With our reporter gene expressed under a constitutive CAG promoter in our system, it was necessary to establish that the compounds in our screen were affecting our AP-1 promoter and not the CAG promoter. When screening agonists, regardless of whether a compound was activating the CAG promoter, a positive hit readout would only be possible if the AP-1 promoter was also activated, in order for the recombinase to remove the stop element. In screening antagonists, however, compounds could inhibit the CAG promoter instead of the AP-1 promoter, resulting in a false-positive antagonist. To address this, a cell line was generated similarly to our reporter, but integrated with a construct consisting of the CAG promoter driving reporter gene expression (Supplementary Fig. 9b). This cell line was then screened against the antagonist reconfirmation library. Results indicated that only three compounds showed decreased reporter gene expression (unaffected by cell viability). However, both our reporter and the classical reporter, which does not rely on a CAG promoter, also indicated inhibition of reporter gene expression, suggesting that the decrease in readout could be due to cell regulation of gene expression unrelated to inhibition of the CAG promoter (Supplementary Fig. 9c).

**Application to genome-wide CRISPR screening.** Due to the digital nature of our reporter's readout, we postulated that it could be highly advantageous for differentiating populations in pooled CRISPR screens. To test this idea, a HEK293 cell line stably expressing Cas9 and our reporter was transduced with a genome-wide sgRNA lentiviral pool. The sgRNA pool was divided into two libraries, CP1 and CP3, to accommodate the number of distinct sgRNAs being tested. The cells were then treated with either DMSO, ABA, or ABA/PMA, followed by sorting of GFP+ and GFP− populations. An unsorted population was also kept as a control to verify that all sgRNAs were well represented following transduction. Next-generation sequencing (NGS) was performed to determine which gRNAs were enriched (Fig. 3a). Following compound treatment, we found that a small population of cells treated with only ABA was expressing GFP. This is exemplified in cells treated with the CP1 library, where the DMSO control showed 0.053% of cells in the GFP+ population while the percentage was 0.45% in the ABA condition (Fig. 3b). While still a small population, this suggests that the genes targeted by these gRNAs inhibit AP-1 activity and, when knocked out, result in constitutive AP-1 activation. When cells transduced with the gRNA library were compared to a non-transduced control following ABA/PMA stimulation, a larger population of cells fell into the GFP− population, suggesting that the knocked-out genes falling into this population were needed to activate AP-1 (Fig. 3c).

Analyzing the sequencing data, a redundant siRNA activity (RSA) metric was used to assess the significance of a gene falling into the GFP+ (RSA-up) or GFP− (RSA-down) population[27]. The more negative the RSA score, the more significant the gene was considered to be. To determine positive regulators of our PKC-induced AP-1 pathway, the GFP− population was of interest, and we compared the RSA-down scores of cells treated with ABA/PMA, with those treated with DMSO or ABA. Here, we found several hits, including *FOS*, *PRKCD*, *RASGRP1*, *ELK1*, and *MAPK1*, all known to be associated with the AP-1 pathway[18,28–31]. Conversely, in identifying negative regulators, we focused on genes falling into the GFP+ population. Using RSA-up values, we were interested in which genes scored

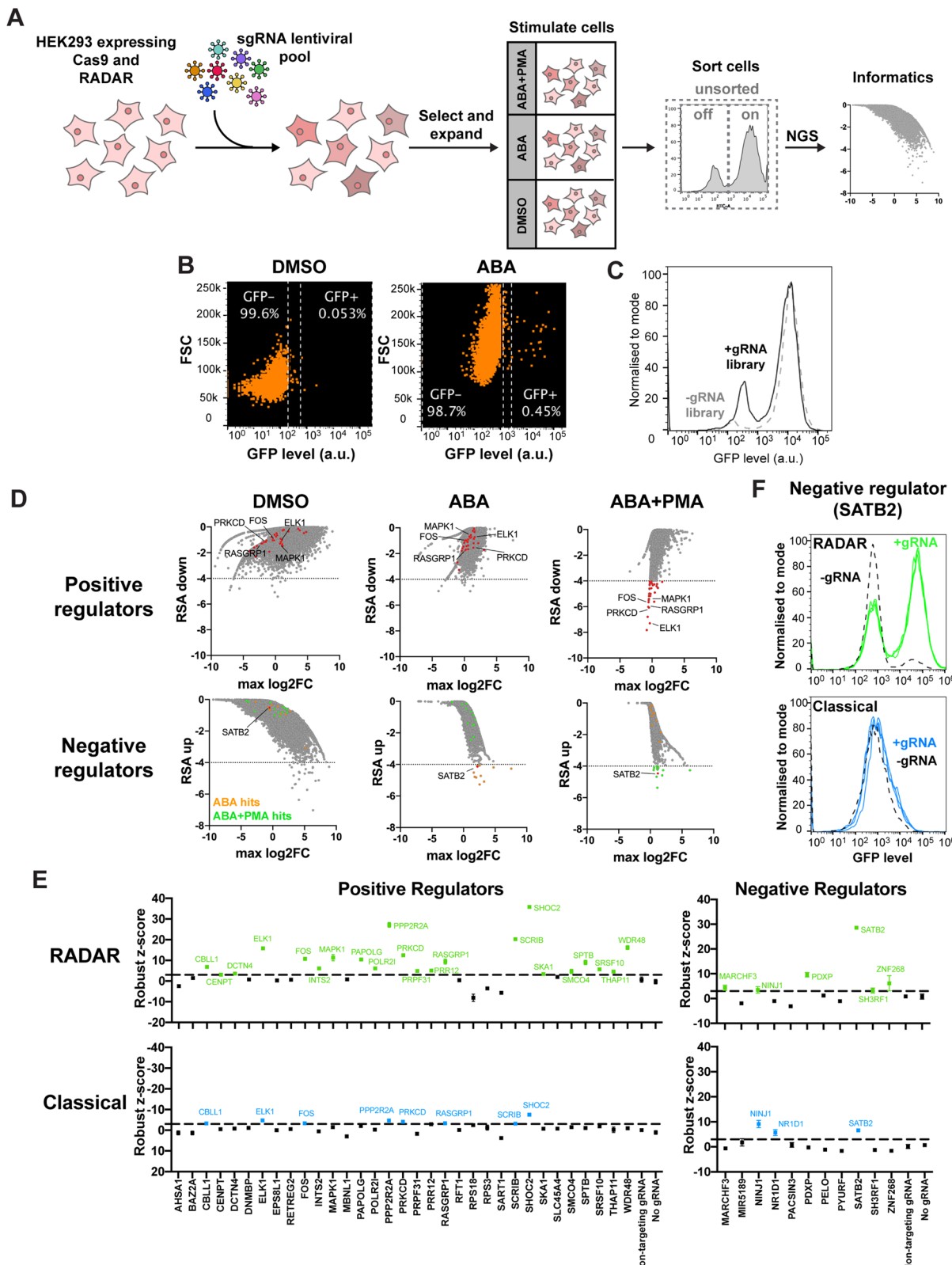

prominently in the ABA condition but not in the DMSO condition, as this would inform us on which genes would result in constitutive AP-1 activity when knocked out. Of these gene hits, only *SH3RF1* had a known STRINGdb association with AP-1, with its encoded protein, believed to act as a scaffold for the c-Jun pathway, though its regulatory role in the AP-1 pathway is unclear (Supplementary Fig. 10). While many gene hits were in

this condition, most of them were not recapitulated in the ABA/PMA condition. However, considering the strong GFP readout upon stimulation with PMA, it is unsurprising that it would be difficult to detect further increases in GFP levels. Interestingly, *SATB2* did come up as a gene hit in both ABA/PMA and ABA conditions, indicating that this gene knockout could be resulting in intense activation of our pathway (Fig. 3d).

**Fig. 3 Application of RADAR to pooled CRISPR screening. A** Schematic of how the pooled CRISPR screen was performed. **B** GFP levels for cells transduced with the CP1 gRNA library and treated with DMSO or ABA. **C** Comparison of RADAR-expressing cell activity (as measured in GFP levels) when treated with or without CP1 gRNA library. **D** RSA-down and RSA-up values for genes in CP1 and CP3 gRNA libraries, plotted against maximum $\log_2$(fold change). Positive regulator gene hits scoring −4 and below are highlighted in red, with example names displayed for genes associated with the PKC-induced AP-1 pathway. Negative regulator gene hits scoring −4 and below are displayed separately for ABA (orange) and ABA+PMA (green) conditions. **E** Robust z-scores of reporter activity, for genes included in the reconfirmation screen. Activity for RADAR was measured in terms of %GFP− cells for positive regulator screen, and %GFP+ cells for negative regulator screen. Activity for classical reporter was measured by GFP mean values for both positive and negative regulator screens. Dotted line indicates 3 standard deviations away from the median negative control activity ($n = 3$, mean ± s.d.). **F** Flow cytometry data for cells expressing either reporter, and with or without gRNA targeting *SATB2*. RADAR and classical reporter-expressing cells were treated with ABA/PMA or PMA, respectively, to activate the AP-1 pathway, then GFP levels measured.

To validate hits from the initial screen, individual genes were knocked out in our reporter cell line, and cells were then treated again with the three drug conditions (DMSO, ABA, ABA/PMA). Given the digital nature of our reporter, percentages of GFP+ or GFP− cells were compared to negative controls (non-targeting gRNA and cells with no knockout) by robust z-scoring, and used to analyze reporter activity. Genes with activity > 3 standard deviations from the median negative control activity were considered hits (as per the 68–95–99.7 rule, 99.73% of the values should fall within three standard deviations). Results from the reconfirmation revalidated the known AP-1-associated genes (*FOS*, *PRKCD*, *RASGRP1*, *ELK1*, and *MAPK1*) and others, including *CBLL1*, *PAPOLG*, *PPP2R2A*, *SCRIB*, *SHOC2*, *SPTB*, and *WDR48*. The knockout of *SHOC2* resulted in a notably larger decrease in reporter activity compared with what was seen with the other positive regulators (35 standard deviations). *SATB2* distinctly reemerged as a strong negative regulator gene, while *MARCHF3*, *NINJ1*, *PDXP*, *SH3RF1*, and *ZNF268* were also potential negative regulator genes.

To compare how our reporter would fare against a classical reporter when evaluating gene knockouts, the reconfirmation was additionally performed against a Cas9 stable line expressing a classical AP-1 reporter. As a measure of reporter activity, mean GFP levels were quantified, similarly compared by robust z-scoring, and the same >3 standard deviations boundary applied. The classical reporter was able to pull out most of the known genes related to our pathway of interest (*FOS*, *ELK1*, *PRKCD*, *RASGRP1*), except for *MAPK1*. Overall, it identified fewer positive regulators than RADAR. *SATB2* was also detected as a negative regulator gene, along with *NR1D1* and *NINJ1*, though the change in activity recorded was not as marked as RADAR (6 standard deviations compared to RADAR's 28) (Fig. 3e). Furthermore, flow cytometry data exhibit the digital behavior of our reporter, with distinct "on" and "off" populations, for which gates can be easily chosen when running a pooled CRISPR screen. This is unlike the classical reporter, which shows a considerable overlap between the two populations (Supplementary Fig. 11a). In the case of SATB2, flow cytometry data exhibit how RADAR clearly captures this negative regulator that has a subtle effect on the classical reporter (Fig. 3f). Upon staining both reporter cell lines for Cas9, we thereafter found that the levels of Cas9 in our RADAR-expressing line were considerably lower than that of the classical reporter (Supplementary Fig. 11b). This was likely due to variation when generating the monoclonal lines, and would have affected the editing efficiency of our reporter. However, it should be noted that despite this, our system was still able to outperform the classical reporter. It also suggests that RADAR could potentially be used in cells that may not tolerate high Cas9 expression levels.

Delving into the genes identified in the CRISPR screen, we were interested in understanding how some of them could be related to our PKC-induced AP-1 pathway. *SHOC2* and *SCRIB* were also detected by both reporters and their corresponding

proteins are known to form a complex that can negatively and positively regulate the ERK pathway[32]. As the ERK pathway is upstream of AP-1 activation, this could explain their identification as positive regulators. *PPP2R2A* was also recognized by both reporters and encodes a regulatory subunit of protein phosphatase 2A, which has been shown to target various proteins in the PKC-induced AP-1 pathway, such as ERK-related proteins and c-Jun[33]. *WDR48* was detected only by our reporter and encodes a regulator of deubiquinating complexes, for which knockdown experiments have led to an accumulation of PHLPP1, which dephosphorylates and inactivates PKC, indicating its connection to our pathway[34,35]. For the remaining genes such as *PAPOLG*, *SPTB*, and *CBLL1* that encode a poly(A) polymerase, cytoskeletal protein, and E3 ubiquitin ligase, respectively, no apparent link to AP-1 is known to the best of our knowledge.

A prominent negative regulator hit identified by RADAR was *SATB2*. SATB2 is involved in transcriptional regulation and chromatin remodeling, and mutations in this gene have been linked to brain development and the causation of oral clefts[36–38]. There is no documented association between SATB2 and the AP-1 pathway. However, SATB2 has been shown to inhibit ERK5 activity in colorectal cancer cells[39]. As it has been shown that ERK5 can induce c-Fos and c-Jun[40,41], it is reasonable that SATB2 could be a negative regulator of AP-1, though it has not yet been directly linked to the pathway. Strangely, both reporters identified NINJ1 as a negative regulator, despite its link to AP-1 activation[42]. Other potential negative regulator hits identified by our reporter were *MARCHF3*, *PDXP*, and *ZNF268*. These encode for a ubiquitin ligase, cytoskeleton-related phosphatase, and transcriptional repressor, respectively[43]. In contrast, the classical reporter pulled out *NR1D1*, which encodes a transcriptional repressor[44]. However, none of these genes have clear associations with AP-1.

Similar to the compound screens, we wanted to verify that the gene knockouts were affecting our AP-1 promoter, and not the constitutive CAG promoter within our system. It is possible that the gene knockout could be affecting the expression of GFP through the CAG promoter. As such, we tested our CAG control in a Cas9 stable line with our reconfirmation panel. Results indicated robustly that none of the gene knockouts were influencing GFP expression through CAG (Supplementary Fig. 12).

## Discussion

Given the importance of cell-based screening for basic research and drug development, much effort has been devoted to improving this assay. We have witnessed explosive progress in methods to efficiently perturb the cells, such as through robotics and discovery of new genome-editing enzymes. Coupling genetic perturbation with single-cell RNA sequencing has enabled high-dimensional analysis of the perturbation. However, these high-dimensional analyses greatly reduce the scalability of the assay. Advances in a transcription reporter system, which remain one of

the most important forms of readout in screening experiments, fall behind compared to the progress made with perturbation methods. Here, we present RADAR—a digitizer circuit that provides memory, increased sensitivity, a large dynamic range, and a digital readout. As a proof of principle, we applied our system to the important AP-1 pathway and tested it with different high-throughput screening methods. We have shown that our reporter was also able to pull out weak agonists in a compound screen and provide a clear distinction between "on" and "off" populations in a pooled CRISPR screen.

The digital nature of this reporter is likely attributed to its design and method of integration. As the reporter is introduced by CRISPR/Cas9 knock-in at the AAVS1 locus, this limits the number of copies that are integrated into the genome. Compared to lentiviral or transient introduction, where numerous copies are integrated, site-specific knock-in lowers the copy number, even when factoring in possible off-target integrations. This could be the reason for the low background activity observed with the reporter, which is also seen with the classical reporter that is integrated in the same manner. When several copies are expressed in the cell, as was observed when the system was introduced by transient transfection, background levels start to increase when cells were treated with ABA alone. This low background activity, compounded with the strength of the CAG promoter, could be the cause of the digital readout.

In compound screens, the larger dynamic range of our system led to a clearer dose response that facilitates identification of drug candidates. Interestingly, our compound screen also showed that the longer cells were incubated in drug and recombinase dimerizer, the more sensitive the reporter could be. A possible reason for this is that the cells may be accumulating more recombinase protein over time, thus raising the probability of the transcription terminator being excised by the protein. In addition, cells that had switched "on" would continue to divide, and since the removal of the transcription terminator is ingrained in the genome of these cells, the newly divided cells would also express the reporter gene. Both of these occurrences would lead to an increase at the bulk readout level. Though not exemplified with our compound screen, the dependency of recombinase dimerization on ABA also introduces the potential to screen compounds that indirectly activate a pathway of interest. This could be done by only treating cells with ABA after the compound has been washed out. It should be noted, however, that the inclusion of a split recombinase can also introduce complications when screening antagonists, as the compounds introduced may interfere with its dimerization. One possible way to circumvent this in future screens would be to cross-verify antagonist candidates with RADAR cell lines that incorporate distinct CIP systems.

The digital nature of RADAR also improves the confidence in identifying candidates in CRISPR screens. Traditionally in CRISPR screening, a percentage of the lowest and highest reporter-expressing cells are selected as the positive and negative regulators, respectively. However, this percentage is arbitrarily chosen, resulting in gates that can be too lenient or stringent. Furthermore, by selecting this somewhat arbitrary population of cells, there is an increased chance of false positives or missed candidates. The digital nature of our reporter helps to identify which cells fall in the reporter "high" and "low" populations and thus circumvent this issue.

Both our high-throughput screens were able to identify several compounds and genes related to our AP-1 pathway, some of which with ties to AP-1 or PKC not previously known. While many of the compounds we identified had previously been known to target PKC, multiple agonists were also found targeting less obvious genes, such as CMA and EDNR. Our CRISPR screen was also able to identify other genes with less apparent links, such as WDR48, SHOC, and SCRIB, thus providing a foundation for future investigations.

A particularly interesting negative regulator that emerged from our pooled CRISPR screen was the gene SATB2, which had significant RSA-up scores in both ABA and ABA/PMA conditions. While its link to AP-1 could be explained through the ERK5 pathway, the notable aspect of this finding was the magnitude of change in reporter activity above PMA induction, which was evident with RADAR and more weakly with the classical reporter. While other negative regulator candidates indicated a fold change of 1.5–3 in reporter activity, SATB2 had a much higher 5-fold increase. A clear association between SATB2 and AP-1 is not yet known, and the results from our CRISPR screen suggest the former could be a potent negative regulator of the latter. Given the interest in developing AP-1 inhibitors for treating cancer and inflammatory diseases such as arthritis, SATB2 could be a consequential target to pursue[45,46].

The importance of the AP-1 pathway is well documented, yet except for a few small-scale shRNA screens using luciferase as the output[47,48], there is no genome-wide shRNA or CRISPR screen on the AP-1 pathway. We postulate that the weak activity of the AP-1 promoter is responsible for the lack of a genome-wide screen. Using luciferase can increase the reporter sensitivity, but it is not compatible with the more scalable pool-based genetic screens. Our RADAR system has addressed this sensitivity issue, thus enabling the first genome-wide screen of the AP-1 pathway.

In its current design, our reporter provides feedback on a single pathway, which can be beneficial as its simplicity can make results easier to deconvolute. However, we have shown its compatibility with other promoters, such as WNT, NF-kB, and CAGA12. In addition to the advantages of memory and sensitivity, the recombinase circuit could theoretically be made more complex to factor in more than one pathway-sensitive promoter, allowing for the screening of multiple pathways simultaneously. This can be used to avoid counter-screening, which can save time, labor, and other resources. The circuit could also be designed to detect serial activation of pathways (e.g., recombinase-driven readout depends on the prior expression of a distinct recombinase), which would allow for the screening of compounds that lead to differentiation. The advantage of memory could also be used in microscopy to visualize how cells activate, whether it be the compound directly activating cells and switching them on in a random fashion, or if the compound induces cells to secrete factors that switch on neighboring cells in a growth ring-like manner. By incorporating these synthetic biology circuits into cells, the screening process could be made more effective and efficient, and the capabilities of cell reporters much expanded.

Overall, RADAR has proven to improve both compound screening and pooled CRISPR screening processes. The monetary and time cost of running several compound screens can be mitigated by running a single screen using our memory-retaining reporter. Furthermore, we have shown that our reporter's sensitivity can be increased with longer incubation periods of the reporter cells with the compound, suggesting weak agonist candidates can be identified. The large dynamic range of our reporter is evident in our high-throughput screen, where we have shown that our signal window is favorable for bulk luciferase readout even when scaling down to 1536-well plates. The ability of our technology to report in a digital manner also removes the arbitrary factor of gating cells in pooled CRISPR screens, and can potentially identify genes previously not known to be associated with target pathways. This could be incredibly beneficial for expanding our understanding of biological pathways and finding new genes for drug targets.

## Methods

**Plasmids**. Reporter constructs were introduced into the AAVS1 site using vectors flanked on both sides by homology arms to the AAVS1 locus. The RADAR vector encoded both recombinase and reporter gene cassettes, separated by a P2A sequence. The stop element incorporated into this vector consisted of a neomycin/kanamycin resistance gene, followed by a SV40 polyA signal. Both RADAR and classical reporters were followed by a selection cassette consisting of a PGK promoter driving expression of a zeocin resistance gene and blue fluorescent protein (BFP). Primers used to clone the RADAR vector are listed in the Supplementary Table. These vectors were transfected into cells along with distinct plasmids encoding Cas9 and an AAVS1-targeting gRNA (sgRNA: 5′-GGGCCACTAGG-GACAGGAT-3′). When the reporter was integrated through the piggyBac method, a distinct vector was used with core insulators flanking the reporter. Cells were transfected concurrently with a separate vector encoding the piggyBac transposase.

**Cell culture**. The reporter cell lines consisted of human embryonic kidney cells (HEK293T; Thermo Fisher #R70007) stably expressing Cas9 that were integrated with respective reporters at the AAVS1 site. All HEK293T derived cell lines were passaged in DMEM supplemented with 10% fetal bovine serum (Seradigm, 1500-500), 2 mM GlutaMAX (Gibco, 35050-061), 100 UI/mL penicillin, 100 μg/mL streptomycin (Gibco, 15140-122), and 1 mM sodium pyruvate (Gibco, 11360-070) and maintained at 37 °C and 5% $CO_2$. Media also contained 5 μg/ml blasticidin (Gibco, A11139-03) to select for Cas9-expressing cells and 400 μg/ml zeocin to select for reporter-expressing cells. Cells were in selection media for a minimum of 10 days prior to testing in assays.

**DNA transfection of HEK293T cells**. For transient transfection, AAVS1 knock-in, and piggyBac integration, plasmids were transfected into cells using a poly-ethylenimine (PEI) method. Cells were plated onto a 6-well plate prior to transfection such that they were 50–70% confluent on the day of transfection. 1.2 μg DNA was added to 120 μl 0.15 M sodium chloride (NaCl; Fisher Scientific), then mixed with 241 μl of a PEI-NaCl mixture consisting of 222 μl 0.15 M NaCl and 19 μl 0.323 g/L PEI (Polysciences, 23966-2). The solution was incubated at room temperature for 10 min, and added dropwise to cells.

**Reporter stimulation assays**. To test the activity of the AP-1 RADAR reporter, 100 μl cells were plated at a concentration of 300,000 cells/ml in 96-well plates in triplicate ($n = 3$) and concurrently treated with combinations of 100 μM abscisic acid (ABA; Gold Biotechnology, A-050-500) and 50 ng/ml phorbol 12-myristate 13-acetate (PMA; Thermo Fisher, BP685-1). Cells were then incubated for 2 days at 37 °C and 5% $CO_2$ and either run on flow cytometry to measure GFP levels (gating strategy as in Supplementary Fig. 13), or lysed in a luciferase assay using Bright-Glo (Promega, E2650) and signal measured on an EnVision plate reader (100 ms). Classical reporter assays were run similarly, but without ABA treatment and only incubated for 1 day in small molecule due to the following loss of reporter gene expression. For WNT, NF-kB, and CAGA12 reporter cells, PMA was replaced with the following agonists and concentrations, respectively: 50% WNT-conditioned media, 7.5 ng/ml TNFα (ProSci, 91-006), 2 ng/ml TGFß (R&D Systems, 240-B).

**Antibody staining**. Cas9 levels were determined through intracellular staining. To fix cells, 1.5–3 million cells were washed with PBS, and resuspended using ice-cold 1% formalin (Thermo Fisher, 28906). Cells were incubated on ice for 15 min, then washed again and resuspended in 200 μl PBS. 1800 μl ice-cold methanol was added while gently shaking, followed by a 30 min incubation on ice. Cas9 was stained using an anti-Cas9 primary antibody (Cell Signaling, 14697, 1:200), and an anti-mouse Alexa Fluor 488 secondary antibody (Thermo Fisher, A11017, 1:1000).

**Compound screen**. HEK293T Cas9 cells expressing the reporters were plated at 5 μl cells/well in white 1536-well plates (Greiner, 789173-A) at 160,000 cells/ml and incubated overnight at 37 °C and 5% $CO_2$. The compound library in 1536-well plates (Labcyte, LP-0400-BC) was then added to cells using an Echo liquid handler, with each well receiving a single compound and each compound tested at 8 doses (final concentrations of 0.0126, 0.0398, 0.126, 0.399, 1.26, 3.99, 12.6, 40 μM). Each compound and dose condition was tested in replicate. Cells were then incubated in compound for 5 h at 37 °C following stimulation by 100 μM ABA for agonist screens, or 100 μM ABA and 50 ng/ml PMA for antagonist screens. Each plate tested contained positive and negative controls. Positive control wells in agonist screens were treated with ABA/PMA, while antagonist screens with ABA. Negative control wells in agonist screens were treated with ABA, while antagonist screens with ABA/PMA. RADAR cells were then incubated for 2 days at 37 °C and 5% $CO_2$ (or longer if indicated), and classical reporter cells for 1 day at 37 °C and 5% $CO_2$. Cells were then lysed in a luciferase assay (Promega, E2650) or cell viability assay (Promega, G7570) and luminescence read on a ViewLux imager (100 ms). All RLU measurements were normalized to the median of the negative controls on their respective plate prior to analysis.

Results from the MoA box compound screen were analyzed using the Novartis "Helios" data analysis software. Compounds were considered hits if the normalized RLU indicated a dose response to increasing compound concentration. Compounds selected as agonist hits had dose–response curve fit types that were categorized by

the software as non-constant (e.g., parametric). As antagonist hits were more numerous, requirements were more stringent and compounds were considered hits if they had a parametric curve fit type without any RLU values above basal levels. Antagonist hits also required at least 3 doses with normalized RLU values below −50%, or the highest two doses below −50% with the highest dose below −80% (to account for less potent antagonists). Results from the reconfirmation screens were analyzed on the Spotfire analytics software. Compounds were individually evaluated for dose–response behavior. In the agonist screens, dose–response curves were qualitatively assessed and considered hits if both duplicates of at least 3 doses had normalized RLU values above basal levels, or if the highest two doses indicated increasing RLU values (to account for less potent agonists). For antagonists, dose–response curves were compared with cell viability curves to screen for potential decreases in RLU values due to cell toxicity. Remaining compounds with clear dose–response curves were considered hits. Genes associated with compound hits were searched on the STRING database v11.0 to map known and predicted protein–protein interactions[49].

**Packaging of lentivirus for pooled CRISPR screen**. Lentivirus encoding the gRNAs libraries were generated by transfecting HEK293T cells. Cells were plated in 5-layer CellSTACKs (Corning) with a total of 210 million cells in 1 L DMEM supplemented with 10% FBS and 1X non-essential amino acids, then incubated overnight at 37 °C and 5% $CO_2$. Per CellSTACK, 510.3 μl TransIT (Mirus, MIR2700) and 18.4 ml Opti-MEM (Invitrogen, 11058021) were mixed and incubated for 5 min at room temperature. 75.6 μg DNA library and 94.5 μg lentiviral packaging plasmid mix (Cellecta, CPCP-K2A) were then added to the mixture and incubated for 15 min at room temperature. The transfection mixture was then added to 1 L media and used to replace the HEK293T cell media. Cells were incubated for 1 day at 37 °C and 5% $CO_2$ and then replaced with 336 ml fresh media and returned to 37 °C and 5% $CO_2$. Three days following this, the viral supernatant was collected and filtered through a 0.45 μm cellulose acetate filter (Corning, 430516). Viral supernatant was stored at −80 °C prior to use in screening.

**Genome-wide pooled CRISPR screen**. HEK293T Cas9 cells expressing the RADAR system were transduced with a pooled sgRNA library in replicate. The genome-wide sgRNA library (based on a previously described library[8,50]) contained ~5 sgRNAs per gene, and was divided into two sub-pools (CPOOL1 and CPOOL3), each containing ~45,000 sgRNAs. It consists largely of published sgRNA sequences[51] with a small portion (< 2%) of proprietary sgRNAs. The public sgRNA sequences and sequences for the screen hits are provided in Supplementary Data 5. Cells were transduced to obtain a minimum of 1000-fold representation of each sgRNA. Reporter-expressing cells were plated in 5-layer CellSTACKs and infected at 0.5 MOI, with virus supplemented with 5 μg/ml polybrene. 268 million cells were infected in 4 5-layer CellSTACKs, with 2 flasks equal to one replicate. The next day, viral media was removed and replaced with media containing 2 μg/ml puromycin (Gibco, A11138-03). Cells were selected for at least 2 weeks with puromycin to ensure sgRNA integration. Transduced cells were treated with DMSO, ABA (100 μM), or ABA/PMA (50 ng/ml) and incubated at 37 °C and 5% $CO_2$ for 2 days prior to sorting. Cell populations were gated by GFP+ and GFP− populations, with an unsorted population as a control. Genomic DNA was then extracted from the sorted cell populations using a QIAamp DNA Blood Maxi Kit (Qiagen, 51194) and sent to the Novartis campus at Basel, Switzerland for next-generation sequencing (NGS).

**Reconfirmation of CRISPR screen**. Based on the regulator hits from the pooled CRISPR screen, lentiviruses were generated encoding the most effective sgRNA for each gene hit (effectiveness determined as the largest magnitude of log2(fold change) among the ~5 sgRNAs targeting that gene). sgRNA sequences were cloned into a pNGx-LV-g003 vector[8] to generate the individual lentiviruses (Supplementary Data 6). HEK293FT cells were plated in 6-well plates (Greiner, 657160) for a 50–70% confluency on the day of transfection. For each sgRNA, 1.5 μl TransIT and 55.1 μl Opti-MEM were mixed and incubated at room temperature for 5 min. 0.23 μg of the sgRNA-encoding DNA plasmid and 0.28 μg lentiviral packaging plasmid mix were then added to the mixture and incubated for 15 min at room temperature before adding dropwise to a single well. Cells were incubated at 37 °C and 5% $CO_2$ for 1 day, then media removed and replaced with 1 ml fresh media per well. After 3 days, viral supernatant was collected and spun at 300 × g for 5 min to remove cell debris. Viral supernatant was frozen at −80 °C prior to use. Cells stably expressing Cas9 and the reporter were then infected with these viruses separately by first plating cells in a 96-well plate (Corning, 353072) with 70 μl cells/well at 300,000 cells/ml. Cells were incubated at 37 °C and 5% $CO_2$ overnight, then media replaced with 50 μl fresh media supplemented with 5 μg/ml polybrene and 17 μl viral supernatant (incubated 15 min at room temperature prior to addition to cells). Media was replaced the following day with fresh media containing 2 μg/ml puromycin and cells selected for a minimum of 14 days. RADAR and constitutive CAG control-expressing cells were treated with DMSO, ABA, or ABA/PMA and incubated for 2 days, while classical reporter-expressing cells were treated with DMSO or PMA and incubated for 1 day. GFP levels were then measured by flow cytometry.

## Statistical analysis

*RSA scoring.* NGS data from the pooled CRISPR screen were analyzed using the RSA algorithm, which calculates the probability of a gene hit incorporating the results from multiple sgRNAs per gene[27]. It is a probability-based method that favors gene hits with multiple active sgRNAs over a single very active sgRNA. Each gene was scored as previously described[52] by calculating log2(fold change) values using DESeq2 based on raw sgRNA counts[53], then using these values as a ranked distribution for RSA scoring (Supplementary Data 7). Genes with RSA-down or RSA-up scores of −4 or below were considered to be positive and negative regulator hits, respectively.

*AC50 calculations for reconfirmation compound screen.* Compound hits for both RADAR and classical reporters in the agonist reconfirmation screen were fitted with a nonlinear regression (sigmoidal, 4PL) using GraphPad Prism 8. The AC50 values of compounds with an unambiguous 95% confidence interval and a fit with $R^2 \geq 0.85$ were plotted in Fig. 2d.

*Signal-to-noise (SNR) calculations.* Signal-to-noise ratios (SNR) were calculated from biological flow cytometry data comparing the geometric mean and standard deviation values of discreet cell populations[54]. We consider the ON state to be samples receiving both pathway- and recombinase-inducing drugs (PMA and ABA for RADAR, PMA alone for classical reporter), and the OFF state samples to be those receiving only recombinase-inducing drug (ABA for RADAR, DMSO for classical reporter). SNR is then calculated using the following formula:

$$\text{SNR} = 20 * \log_{10} \frac{\left| \log_{10} \mu_{g,ON} / \mu_{g,OFF} \right|}{2 * \log_{10} \left( \sigma_{g,ON} + \sigma_{g,OFF} \right) / 2} \tag{1}$$

where $\mu_{g,ON}$ and $\mu_{g,OFF}$ represent the geometric mean of the ON or OFF state populations, respectively, and $\sigma_{g,ON}$ and $\sigma_{g,OFF}$ represent the geometric standard deviation of the ON and OFF state populations, respectively. Note that Eq. (1) can only be used to calculate the SNR of unimodal populations.

## Statistics and reproducibility

Experimental data were collected from 1 to 3 independent experiments. The number of repeated experiments is as follows: three times—Figs. 1a, S2e; twice—Figs. 1h, S2c; once—Figs. 1e-g, S2a-b, S2d, S3. Experiments other than the reporter activity assays consisted of the compound and CRISPR genome-wide screens. Due to the large-scale nature of these high-throughput screens, they were each performed a single time. However, within the compound screen, compounds were tested in replicate, with multiple doses corresponding to each drug tested, and cells in the genome-wide CRISPR screen were transduced to obtain a minimum of 1000-fold representation of each sgRNA.

## Reporting summary

Further information on research design is available in the Nature Research Reporting Summary linked to this article.

## Data availability

The data that support the findings of this study can be found in the associated supplementary data (Supplementary Data 1–7) and source data. Protein–protein interaction analysis was performed on STRING (https://string-db.org/) (Version 11.0). CRISPR-Cas9 screen data and chemical screen data are summarized in Supplementary files 1–7. The MoA chemical compound information is available on GitHub - Novartis/MoaBox: A repository of compound-target annotations in support of Systematic Chemogenetic Library Assembly. The proprietary guideRNA sequences provided by Novartis used in this work can be made available for research purposes subject to a Material Transfers Agreement. Requests should be made to M. Hild. All other data are available from the corresponding author upon reasonable request. Source data are provided with this paper.

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

## Acknowledgements

W.W.W. acknowledges funding from the BU/NIBR Bridge Award, NIH (1DP2CA186574, 1R01GM129011-01, R01EB029483), NSF Expedition in Computing (1522074), NSF CAREER (162457), and NSF BBSRC (1614642). We thank Dr. Todd Blute from the Boston University Proteomics & Imaging Core Facility, as well as Dr. Akos Szilvasi, Steven Paula, and Duncan Nunes from the Novartis Institutes for Biomedical Research (NIBR) for flow cytometry assistance. We also thank the Next Gen- eration Sequencing team at NIBR for assistance in CRISPR screen sequencing, as well as Peter Aspesi and Dr. Eugen Lounkine for their assistance and contributions to the high-throughput compound screens. We thank Dr. Jeremy Jenkins and Dr. Douglas Auld for their suggestions on the manuscript.

## Author contributions

N.M.W designed and generated genetic constructs, performed experiments, analyzed the data, and generated figures. E.F. guided and assisted in performing compound and CRISPR screens. F.D.S. helped analyze CRISPR screen data. J.H.L. carried out SNR calculations. W.W.W and M.H. conceived the project and analyzed the data. All authors commented on and approved the paper.

## Competing interests

The authors declare the following competing interests: W.W.W. is a co-founder and shareholder of Senti Biosciences, and received research support from Senti Biosciences. N.M.W. is a current employee of Ginkgo Bioworks. All other authors declare no competing interests.
