## [Peer Review File · Nature Communications]

Reviewers' Comments:

Reviewer #1:

Remarks to the Author:

Summary:

In this manuscript, the authors aim to build and test a cell-based transcriptional reporter for the purpose of reporting on the underlying biology and using information gleaned from these studies to identify drug candidates that can modulate the disease phenotype. As proof of concept the authors use their recombinase-based analog-to-digital reporter (RADAR) to monitor the activator protein (AP-1) pathway through the induction of protein kinase C (PKC) signal transduction. The technology described in this study is interesting and can be broadly applied. However, in its current form, the manuscript requires several points to be addressed in order to demonstrate the value of the proposed technology and application.

Comments for the authors:

1. The introduction takes a while to get to the point. An introduction that focuses on the specific background needed to understand the rationale for building a system made up of recombinases would improve this. For example, combining some of the current paragraphs (they are redundant) and adding one on the protein kinase C signal transduction pathway (how it relates to cancer) and how the AP-1 pathway connects to it would be useful for the reader to fully grasp the importance of this novel tool. Additionally, a short background on recombinases would help get the reader caught up with the technology used. Finally the last paragraph of the introduction is vague and unclear on what the authors will do. I suggest describing what large-scale compound HTS and pooled CRISPR screens are and why they are being chosen for the study. As screening tools for enhancing signals from the RADAR system?

2. The language of the sequence between the *frt* sites in the RADAR system is unconventional. Typically transcription terminator sequences (denoted a T in the schematic) are used in prokaryotes. However, this is not the convention for eukaryotes. What is the terminator sequence used? Moreover, in the field, such a sequence is typically referred to as a 'stop element' that is located upstream of a gene (here the reporter) and prevents transcription of the reporter until it is removed by the recombinase.

3. While I understand that PMA is used to activate the AP-1 pathway, how do the concentrations in the results compare to physiological levels required to activate the AP-1 pathway *in vivo*? In other words, do the concentrations used to test the RADAR system make physiological sense? How sensitive is the AP-1 promoter? Does this promoter give true readouts to phenotypic outcomes?

4. The authors mention that they use a classical AP-1 reporter to compare it to the RADAR system. What is the classic AP-1 reporter? How does it work? How is the RADAR system different in design and expected outcomes? Related to this, in figure 1e, why do the authors believe that the GFP expression is decreasing in the classic model and increasing in the RADAR system?

5. In the application of RADAR to screen compounds, it is unclear why the authors are looking for inhibition of the AP-1 related hits since the RADAR system was designed to be an AP-1 sensitive promoter – in figure 1 the system was introduced as the detection of AP-1, which would turn on the transcription of FlpO (so activation of the promoter) that ultimately cuts out the stop element and turns on the reporter. How are the authors using this same system to inhibit the AP-1 promoter for drug screening? Maybe I missed this explanation but this is a confusing point in the manuscript that may warrant more description.

6. More details on the sgRNA pools would be informative to better understand the results in figure 3. What are they designed to? The authors state that the library is based on a "previously described library" but there is no reference for this statement.

7. What is the rationale for knocking out the particular genes in the study? Do any of these genes have known redundancy? For example, if one is knocked out, does another increase its expression? Some background on these genes would be helpful to fully understand the study.

8. More description in the figure legends on what each sample is would be helpful. For example, the inducible promoter in the case of figure 1a should list the sensitive promoters used in the study in the legend. In figure 3A the informatics figure has blue and pink dots. What are the pink dots?

9. Why are there bold terms in the introduction?

10. This study is not well organized and lacks a cohesive hypothesis, which makes it difficult to follow the scientific direction. The authors state that they want to build a tool to obtain accurate and sensitive measurements of various cellular phenotypes/pathway activities, however, the execution of this goal is muddled with less-focused experiments. More details on the rationale for each study and how it relates to their goal of reporting on various cellular phenotypes and pathways would significantly increase readership and impact.

Reviewer #2:

Remarks to the Author:

One important approach in drug discovery to uncover new targets and drug candidates is phenotypic cell-based screening. However, many phenotypic outputs do not allow to identify nuanced perturbations. One such readout is transcription modulation, which is phenotypically assessed using transcription reporters in response to pathway activation or inhibition. Very often these transcription reporters are weak and/or transient which makes it difficult to detect genes or compounds with small modulatory effects. Therefore, the authors have developed a recombinase based analog to digital reporter system, named RADAR. Although this system requires extensive optimization and while its versatility towards application in different cell lines is uncertain, this work forms a significant advancement in the field. The manuscript is well written and the experiments seem thorough and sound.

Major comments:

The study would benefit from more specific information on the hits identified with RADAR. For example, results and information on the 5 agonists and 27 antagonists identified from the 3494 compound library is missing. This is especially important because these are used to validate the method.

There is very limited information on the genome wide CRISPR library used. The sequence of the guides in the library should be given in a table as this seems a not so commonly used CRISPR library.

For the reconfirmation of the CRISPR screen authors used the most effective sgRNA for each hit. The hit validation would benefit if it is performed with a different guide sequence targeting the same gene instead of the same guide sequence used in the original screen. This is especially important because many hits were previously unknown to be associated with AP-1 and others may want to follow-up on these hits.

Minor comments:

Page 5. In the paragraph 'Application to compound screening' (second sentence) it is unclear which compound is referred to.

Number of replicates and number of independent experiments used to obtain the results should be indicated in the figure legends

How did the others do hit calling? When was a compound called a positive hit.

Antagonists may be false positive if they interfere with the split recombinase, this should be discussed.

FigS6. What were the parameters for hit calling? C988 and C3194 seem to show very low activity, but were identified as hits. Also, do these hits really have a statistically lower AC50 for RADAR as compared to classical?

FigS12. Was the GFP intensity taken into account or only a cut off for GFP+ or GFP- cells? Knock out of a gene may decrease GFP expression to a level at which the cells still score positive but the level is decreased, this may influence the results.

REVIEWER COMMENTS

Reviewer #1 (Remarks to the Author):

Summary:

In this manuscript, the authors aim to build and test a cell-based transcriptional reporter for the purpose of reporting on the underlying biology and using information gleaned from these studies to identify drug candidates that can modulate the disease phenotype. As proof of concept the authors use their recombinase-based analog-to-digital reporter (RADAR) to monitor the activator protein (AP-1) pathway through the induction of protein kinase C (PKC) signal transduction. The technology described in this study is interesting and can be broadly applied. However, in its current form, the manuscript requires several points to be addressed in order to demonstrate the value of the proposed technology and application.

Comments for the authors:

1. The introduction takes a while to get to the point. An introduction that focuses on the specific background needed to understand the rationale for building a system made up of recombinases would improve this. For example, combining some of the current paragraphs (they are redundant) and adding one on the protein kinase C signal transduction pathway (how it relates to cancer) and how the AP-1 pathway connects to it would be useful for the reader to fully grasp the importance of this novel tool. Additionally, a short background on recombinases would help get the reader caught up with the technology used. Finally the last paragraph of the introduction is vague and unclear on what the authors will do. I suggest describing what large-scale compound HTS and pooled CRISPR screens are and why they are being chosen for the study. As screening tools for enhancing signals from the RADAR system?

Thank you for your comments. Please find that we have revised our introduction by cutting down on the specified sections, and further developed the discussion on the AP-1 pathway and motivation behind the large-scale screens.

2. The language of the sequence between the frt sites in the RADAR system is unconventional. Typically transcription terminator sequences (denoted a T in the schematic) are used in prokaryotes. However, this is not the convention for eukaryotes. What is the terminator sequence used? Moreover, in the field, such a sequence is typically referred to as a 'stop element' that is located upstream of a gene (here the reporter) and prevents transcription of the reporter until it is removed by the recombinase.

The T in the schematic has been updated to reflect the more conventional stop element denotation. The terminator sequenced used here consisted of a neomycin/kanamycin resistance gene, followed by a SV40 polyA signal. References to the transcription terminator have also been updated to indicate a termination signal (STOP) instead.

3. While I understand that PMA is used to activate the AP-1 pathway, how do the concentrations in the results compare to physiological levels required to activate the AP-1 pathway in vivo? In other words, do the concentrations used to test the RADAR system make physiological sense? How sensitive is the AP-1 promoter? Does this promoter give true readouts to phenotypic outcomes?

The AP-1 promoter that we used is commonly employed to study AP-1 pathway activation. Since AP-1 is involved in a wide-range of processes in healthy and disease states, it would be difficult to pinpoint physiological levels. Furthermore, our RADAR system can improve the sensitivity the AP-1 promoter and remember weak and transient signals. Therefore, our system represents a substantial improvement over the current state of the art.

We agree that it would be interesting to see how PMA-induced activation of AP-1 in our system would compare to a physiological scenario. However, to the best of our knowledge, it is not known how representative this artificial stimulation is of in vivo activation, nor are there murine models that have delved into PMA-induced AP-1 activation.

4. The authors mention that they use a classical AP-1 reporter to compare it to the RADAR system. What is the classic AP-1 reporter? How does it work? How is the RADAR system different in design and expected outcomes? Related to this, in figure 1e, why do the authors believe that the GFP expression is decreasing in the classic model and increasing in the RADAR system?

Thank you for your comments. The classical reporter discussed in the paper refers to the traditional pathway reporter design consisting of a pathway-sensitive promoter driving expression of a gene for readout (e.g. fluorescent protein or luciferase). The constructs in Figure 1B detail the differences between the classical reporter and RADAR. With RADAR, pathway activation results in the expression of the recombinase instead, which makes a permanent change by removing the stop signal preceding the readout gene. This results in a reporter that has memory because once the stop signal is removed, the reporter is perpetually on and does not require continual pathway activation. In Figure 1E, GFP could be decreasing in the classical reporter due to degradation of the initial PMA over time or natural attenuation of PKC signalling pathway. The RADAR system, however, does not require constant activation of the AP-1 promoter to sustain GFP expression due to its memory characteristic. The increase in GFP from day 1 to 4 may be due to the accumulation of GFP that is constitutively expressed in each cell. Please find that we have appended the manuscript to include discussion of these questions.

5. In the application of RADAR to screen compounds, it is unclear why the authors are looking for inhibition of the AP-1 related hits since the RADAR system was designed to be an AP-1 sensitive promoter – in figure 1 the system was introduced as the detection of AP-1, which would turn on the transcription of FlpO (so activation of the promoter) that ultimately cuts out the stop element and turns on the reporter. How are the authors using this same system to inhibit the AP-1 promoter for drug screening? Maybe I missed this

explanation but this is a confusing point in the manuscript that may warrant more description.

Thank you for your comments. In the antagonist compound screens, it is expected that if a particular compound was an inhibitor of the AP-1 pathway, this would prevent the later addition of PMA from activating our AP-1 reporter. For this reason, the compound library was added to the cells a few hours prior to the addition of PMA. We have added an explanation of this to the manuscript to help with the clarification.

6. More details on the sgRNA pools would be informative to better understand the results in figure 3. What are they designed to? The authors state that the library is based on a “previously described library” but there is no reference for this statement.

Thank you for pointing this out. The sgRNA pools combined result in a genome-wide library. Please find that we have updated the manuscript to reflect this. The previously described library is in the papers by Alimov et al. (PMID 30647128) and DeJesus et al. (PMID 27351204) in reference numbers 8 and 49. Please note that the exact number of genes in the screens may differ slightly due to gRNA dropout when the screen is performed.

7. What is the rationale for knocking out the particular genes in the study? Do any of these genes have known redundancy? For example, if one is knocked out, does another increase its expression? Some background on these genes would be helpful to fully understand the study.

The sgRNA library used in this study had genome-wide coverage to determine if we could pull out any AP-1 related genes with our reporter. Therefore, we didn't choose to study any particular genes. We highlighted the genes that were uncovered from our CRISPR screen and provided some background on a few genes that are less known to be associated to the AP-1 pathway, such as SATB2. While powerful, genetic knockout screens are not particularly adept in identifying redundant genes.

8. More description in the figure legends on what each sample is would be helpful. For example, the inducible promoter in the case of figure 1a should list the sensitive promoters used in the study in the legend. In figure 3A the informatics figure has blue and pink dots. What are the pink dots?

Thank you for the feedback, we have updated the legends for clarification. In figure 3A, the informatics figure was hypothetical and solely for schematic purposes. We have updated it to reflect data more specific to our study.

9. Why are there bold terms in the introduction?

The bold terms are for emphasizing key points. Please find the formatting of the introduction changed accordingly.

10. This study is not well organized and lacks a cohesive hypothesis, which makes it difficult to follow the scientific direction. The authors state that they want to build a tool to obtain accurate and sensitive measurements of various cellular phenotypes/pathway activities, however, the execution of this goal is muddled with less-focused experiments. More details on the rationale for each study and how it relates to their goal of reporting on various cellular phenotypes and pathways would significantly increase readership and impact.

Thank you for the comments. Based on your and other reviewers' comments, we have modified the manuscript to enhance its clarity. We will further work with the editors at Nature Communications to improve the manuscript.

Reviewer #2 (Remarks to the Author):

One important approach in drug discovery to uncover new targets and drug candidates is phenotypic cell-based screening. However, many phenotypic outputs do not allow to identify nuanced perturbations. One such readout is transcription modulation, which is phenotypically assessed using transcription reporters in response to pathway activation or inhibition. Very often these transcription reporters are weak and/or transient which makes it difficult to detect genes or compounds with small modulatory effects. Therefore, the authors have developed a recombinase based analog to digital reporter system, named RADAR. Although this system requires extensive optimization and while its versatility towards application in different cell lines is uncertain, this work forms a significant advancement in the field. The manuscript is well written and the experiments seem thorough and sound.

Major comments:

The study would benefit from more specific information on the hits identified with RADAR. For example, results and information on the 5 agonists and 27 antagonists identified from the 3494 compound library is missing. This is especially important because these are used to validate the method.

Please find this information in the Supplementary Tables document.

There is very limited information on the genome wide CRISPR library used. The sequence of the guides in the library should be given in a table as this seems a not so commonly used CRISPR library.

Thank you for your comments. We have updated our manuscript to provide references to the literature that contains more details on the sgRNA sequences used in the library. While a small portion of the sgRNAs (<2%) were designed internally at Novartis and cannot be shared, the majority of the sgRNA designs are available to the public (please refer to reference 8 by DeJesus et al.) For the sequences that are not available, the same reference also provides the criteria for how these sgRNAs were designed.

For the reconfirmation of the CRISPR screen authors used the most effective sgRNA for each hit. The hit validation would benefit if it is performed with a different guide sequence targeting the same gene instead of the same guide sequence used in the original screen. This is especially important because many hits were previously unknown to be associated with AP-1 and others may want to follow-up on these hits.

Thank you for your comments. In the pooled CRISPR screen, multiple sgRNAs (~5) were included for targeting each gene. The following RSA scoring of positive and negative regulator hits favors multiple active sgRNAs over single very active sgRNAs, so the resulting high scoring hits would have taken different guide sequences into account. While it would certainly be useful to have an additional sgRNA designed, we feel it may not be necessary given the method for CRISPR screen analysis.

Minor comments:

Page 5. In the paragraph 'Application to compound screening' (second sentence) it is unclear which compound is referred to.

Our apologies, the obscurity is due to a typo in the sentence that we have corrected in our revision. As the chemogenetic library consists of compounds that cover a broad range of targets, we had meant to indicate it can be used to study the mechanism of action of various compounds, not a specific one.

Number of replicates and number of independent experiments used to obtain the results should be indicated in the figure legends

Please find the legends updated for the main and supplemental figures.

How did the others do hit calling? When was a compound called a positive hit.

Results from the MoA box compound screen were analyzed using the Novartis "Helios" data analysis software. Compounds were considered hits if the normalized RLU indicated a dose response to increasing compound concentration. Compounds selected as agonist hits had dose response curve fit types that were categorized by the software as non-constant (e.g. parametric). As antagonist hits were more numerous, the requirements were more stringent and compounds needed to have a parametric curve fit type and could not have any RLU values above basal levels. Antagonist hits also needed at least 3 doses with normalized RLU values below -50%, or the highest two doses below -50% with the highest dose below -80% (to account for potentially less potent antagonists).

Results from the reconfirmation screens were analyzed on the Spotfire analytics software due to the complexity of the various conditions tested. Compounds were individually evaluated for dose response behaviour. In the agonist screens, dose response curves were qualitatively assessed and considered hits if both duplicates of at least 3 doses had normalized RLU values above basal levels, or if the highest two doses indicated increasing

RLU values (to account for less potent agonists). For antagonists, dose response curves were compared with cell viability curves to weed out any decreases in RLU due to cell toxicity. Remaining compounds with clear dose response curves were considered hits. Please find that we have updated our methods section to reflect the specifications for hit calling.

Antagonists may be false positive if they interfere with the split recombinase, this should be discussed.

Thank you for your comment, we have revised the discussion section to include this.

“It should be noted, however, that the inclusion of a split recombinase can also introduce complications when screening antagonists, as the compounds introduced may interfere with its dimerization. One possible way to circumvent this in future screens would be to cross-verify antagonist candidates with RADAR cell lines that incorporate distinct CIP systems.”

FigS6. What were the parameters for hit calling? C988 and C3194 seem to show very low activity, but were identified as hits. Also, do these hits really have a statistically lower AC50 for RADAR as compared to classical?

Please refer to the above for conditions for hit calling. Thank you for identifying the issue with the C988 and C3194 plots in figure S6. While these compound have lower activity, they were considered hits due to the correlation between increasing compound dose and RLU. This is observed more clearly with a more focused axis range, and we have updated figure S6 accordingly to exhibit this. The AC50 values were calculated using a nonlinear regression (sigmoidal, 4PL) on Graphpad, and the fits with an unambiguous 95% confidence interval and $R^2 \geq 0.85$ were plotted in Figures 2d and S6.

FigS12. Was the GFP intensity taken into account or only a cut off for GFP+ or GFP-cells? Knock out of a gene may decrease GFP expression to a level at which the cells still score positive but the level is decreased, this may influence the results.

Thank you for the comment. In our RADAR system, GFP level is not a direct readout of the pathway activity. GFP can either be on the ON or OFF state and the variation between cells is likely due to intrinsic and extrinsic noise from promoters. However, the probability of GFP being on the ON state is related to concentration of the recombinase, which is directly related to the activity of the AP-1 promoter. Therefore, the number of cells in the ON state, as opposed to the level of GFP, is the indication of AP-1 activity.

Reviewers' Comments:

Reviewer #1:

None

Reviewer #2:

Remarks to the Author:

The manuscript is improved, but there is still too little information on the CRISPR library used. In the references the authors refer to there is no list with guide sequences that are in the library. It would be good for the research community to share these sequences in a supplementary table. Also, why do the authors not share the small portion of guides (<2%) designed at Novartis? The legends are now updated indicating the number of samples (n) but it is not clear whether n represents all samples from a single experiment or whether these are data from multiple independent experiments. This reviewer believes it is important to indicate whether the experiment was just done once or whether results were confirmed by a repeat experiment.

REVIEWERS' COMMENTS

Reviewer #2 (Remarks to the Author):

The manuscript is improved, but there is still too little information on the CRISPR library used. In the references the authors refer to there is no list with guide sequences that are in the library. It would be good for the research community to share these sequences in a supplementary table. Also, why do the authors not share the small portion of guides (<2%) designed at Novartis? The legends are now updated indicating the number of samples (n) but it is not clear whether n represents all samples from a single experiment or whether these are data from multiple independent experiments. This reviewer believes it is important to indicate whether the experiment was just done once or whether results were confirmed by a repeat experiment.

Thank you for your comments. The sgRNA library we have used is described in reference 8, and the majority of its sequences are publicly available in reference 50. For simplification, we have updated our supplementary data to include the publicly available sgRNA sequences, in addition to the sequences corresponding to all our CRISPR screen hits. The small portion of guides designed at Novartis unfortunately cannot be shared due to potential consequences for IP. In the figure legends, the number of samples (n) indicates the replicates within the single experiment that is being shown in the figures. Reporter activity assays were performed 1-3 times, achieving similar results. Experiments other than the reporter activity assays consisted of the compound and CRISPR genome-wide screens. Due to the large scale nature of these high-throughput screens, they were each performed a single time. However, within the compound screen, compounds was tested in replicate, with multiple doses corresponding to each drug tested, and cells in the genome-wide CRISPR screen were transduced to obtain a minimum of 1000-fold representation of each sgRNA. This information will be provided under the "Statistics and Reproducibility" section in the Methods.